# On Tensor Train Rank Minimization: Statistical Efficiency and Scalable Algorithm

**Masaaki Imaizumi**
Institute of Statistical Mathematics
RIKEN Center for Advanced Intelligence Project
imaizumi@ism.ac.jp

**Takanori Maehara**
RIKEN Center for Advanced Intelligence Project
takanori.maehara@riken.jp

**Kohei Hayashi**
National Institute of Advanced Industrial Science and Technology
RIKEN Center for Advanced Intelligence Project
hayashi.kohei@gmail.com

## Abstract

Tensor train (TT) decomposition provides a space-efficient representation for higher-order tensors. Despite its advantage, we face two crucial limitations when we apply the TT decomposition to machine learning problems: the lack of statistical theory and of scalable algorithms. In this paper, we address the limitations. First, we introduce a convex relaxation of the TT decomposition problem and derive its error bound for the tensor completion task. Next, we develop a randomized optimization method, in which the time complexity is as efficient as the space complexity is. In experiments, we numerically confirm the derived bounds and empirically demonstrate the performance of our method with a real higher-order tensor.

## 1 Introduction

Tensor decomposition is an essential tool for dealing with data represented as multidimensional arrays, or simply, tensors. Through tensor decomposition, we can determine latent factors of an input tensor in a low-dimensional multilinear space, which saves the storage cost and enables predicting missing elements. Note that, a different multilinear interaction among latent factors defines a different tensor decomposition model, which yields several variations of tensor decomposition. For general purposes, however, either Tucker decomposition [29] or *CANDECOMP/PARAFAC (CP) decomposition* [8] model is commonly used.

In the past three years, an alternative tensor decomposition model, called *tensor train (TT)* decomposition [21] has actively been studied in machine learning communities for such as approximating the inference on a Markov random field [18], modeling supervised learning [19, 24], analyzing restricted Boltzmann machine [4], and compressing deep neural networks [17]. A key property is that, for higher-order tensors, TT decomposition provides more space-saving representation called TT format while preserving the representation power. Given an order-$K$ tensor (i.e., a $K$-dimensional tensor), the space complexity of Tucker decomposition is exponential in $K$, whereas that of TT decomposition

is linear in $K$. Further, on TT format, several mathematical operations including the basic linear algebra operations can be performed efficiently [21].

Despite its potential importance, we face two crucial limitations when applying this decomposition to a much wider class of machine learning problems. First, its statistical performance is unknown. In Tucker decomposition and its variants, many authors addressed the generalization error and derived statistical bounds (e.g. [28, 27]). For example, Tomioka *et al.*[28] clarify the way in which using the convex relaxation of Tucker decomposition, the generalization error is affected by the rank (i.e., the dimensionalities of latent factors), dimension of an input, and number of observed elements. In contrast, such a relationship has not been studied for TT decomposition yet. Second, standard TT decomposition algorithms, such as alternating least squares (ALS) [6, 30] , require a huge computational cost. The main bottleneck arises from the singular value decomposition (SVD) operation to an "unfolding" matrix, which is reshaped from the input tensor. The size of the unfolding matrix is huge and the computational cost grows exponentially in $K$.

In this paper, we tackle the above issues and present a scalable yet statistically-guaranteed TT decomposition method. We first introduce a convex relaxation of the TT decomposition problem and its optimization algorithm via the alternating direction method of multipliers (ADMM). Based on this, a statistical error bound for tensor completion is derived, which achieves the same statistical efficiency as the convex version of Tucker decomposition does. Next, because the ADMM algorithm is not sufficiently scalable, we develop an alternative method by using a randomization technique. At the expense of losing the global convergence property, the dependency of $K$ on the time complexity is reduced from exponential to quadratic. In addition, we show that a similar error bound is still guaranteed. In experiments, we numerically confirm the derived bounds and empirically demonstrate the performance of our method using a real higher-order tensor.

## 2  Preliminaries

### 2.1  Notation

Let $\mathcal{X} \subset \mathbb{R}^{I_1 \times \cdots \times I_K}$ be the space of order-$K$ tensors, where $I_k$ denotes the dimensionality of the $k$-th mode for $k = 1, \ldots, K$. For brevity, we define $I_{<k} := \prod_{k' < k} I_{k'}$; similarly, $I_{\leq k}, I_{k<}$ and $I_{k\leq}$ are defined. For a vector $Y \in \mathbb{R}^d$, $[Y]_i$ denotes the $i$-th element of $Y$. Similarly, $[X]_{i_1,\ldots,i_K}$ denotes the $(i_1, \ldots, i_K)$ elements of a tensor $X \in \mathcal{X}$. Let $[X]_{i_1,\ldots,i_{k-1},:,i_{k+1},\ldots,i_K}$ denote an $I_k$-dimensional vector $(X_{i_1,\ldots,i_{k-1},j,i_{k+1},\ldots,i_K})_{j=1}^{I_k}$ called the mode-$k$ fiber. For a vector $Y \in \mathbb{R}^d$, $\|Y\| = (Y^T Y)^{1/2}$ denotes the $\ell_2$-norm and $\|Y\|_\infty = \max_i |[Y]_i|$ denotes the max norm. For tensors $X, X' \in \mathcal{X}$, an inner product is defined as $\langle X, X' \rangle := \sum_{i_1,\ldots,i_K=1}^{I_1 \ldots I_K} X(i_1, \ldots, i_K) X'(i_1, \ldots, i_K)$ and $\|X\|_F = \langle X, X \rangle^{1/2}$ denotes the Frobenius norm. For a matrix $Z$, $\|Z\|_s := \sum_j \sigma_j(Z)$ denotes the Schatten-1 norm, where $\sigma_j(\cdot)$ is a $j$-th singular value of $Z$.

### 2.2  Tensor Train Decomposition

Let us define a tuple of positive integers $(R_1, \ldots, R_{K-1})$ and an order-3 tensor $G_k \in \mathbb{R}^{I_k \times R_{k-1} \times R_k}$ for each $k = 1, \ldots, K$. Here, we set $R_0 = R_K = 1$. Then, TT decomposition represents each element of $X$ as follows:

$$X_{i_1,\ldots,i_K} = [G_1]_{i_1,:,:}[G_2]_{i_2,:,:} \cdots [G_K]_{i_K,:,:}. \tag{1}$$

Note that $[G_k]_{i_k,:,:}$ is an $R_{k-1} \times R_k$ matrix. We define $\mathcal{G} := \{G_k\}_{k=1}^K$ as a set of the tensors, and let $X(\mathcal{G})$ be a tensor whose elements are represented by $\mathcal{G}$ as (1). The tuple $(R_1, \ldots, R_{K-1})$ controls the complexity of TT decomposition, and it is called a *Tensor Train (TT) rank*. Note that TT decomposition is universal, i.e., any tensor can be represented by TT decomposition with sufficiently large TT rank [20].

When we evaluate the computational complexity, we assume the shape of $\mathcal{G}$ is roughly symmetric. That is, we assume there exist $I, R \in \mathbb{N}$ such that $I_k = O(I)$ for $k = 1, \ldots, K$ and $R_k = O(R)$ for $k = 1, \ldots, K - 1$.

## 2.3 Tensor Completion Problem

Suppose there exists a true tensor $X^* \in \mathcal{X}$ that is unknown, and a part of the elements of $X^*$ is observed with some noise. Let $S \subset \{(j_1, j_2, \ldots, j_K)\}_{j_1,\ldots,j_K=1}^{I_1,\ldots,I_K}$ be a set of indexes of the observed elements and $n := |S| \leq \prod_{k=1}^{K} I_k$ be the number of observations. Let $j(i)$ be an $i$-th element of $S$ for $i = 1, \ldots, n$, and $y_i$ denote $i$-th observation from $X^*$ with noise. We consider the following observation model:

$$y_i = [X^*]_{j(i)} + \epsilon_i, \tag{2}$$

where $\epsilon_i$ is i.i.d. noise with zero mean and variance $\sigma^2$. For simplicity, we introduce observation vector $Y := (y_1, \ldots, y_n)$, noise vector $\mathcal{E} := (\epsilon_1, \ldots, \epsilon_n)$, and rearranging operator $\mathfrak{X} : \mathcal{X} \to \mathbb{R}^n$ that randomly picks the elements of $X$. Then, the model (2) is rewritten as follows:

$$Y = \mathfrak{X}(X^*) + \mathcal{E}.$$

The goal of tensor completion is to estimate the true tensor $X^*$ from the observation vector $Y$. Because the estimation problem is ill-posed, we need to restrict the degree of freedom of $X^*$, such as rank. Because the direct optimization of rank is difficult, its convex surrogation is alternatively used [2, 3, 11, 31, 22]. For tensor completion, the convex surrogation yields the following optimization problem [5, 14, 23, 26]:

$$\min_{X \in \Theta} \left[ \frac{1}{2n} \|Y - \mathfrak{X}(X)\|^2 + \lambda_n \|X\|_{s*} \right], \tag{3}$$

where $\Theta \subset \mathcal{X}$ is a convex subset of $\mathcal{X}$, $\lambda_n \geq 0$ is a regularization coefficient, and $\| \cdot \|_{s*}$ is the overlapped Schatten norm defined as $\|X\|_{s*} := \frac{1}{K} \sum_{k=1}^{K} \|\widetilde{X}_{(k)}\|_s$. Here, $\widetilde{X}_{(k)}$ is the $k$-unfolding matrix defined by concatenating the mode-$k$ fibers of $X$. The overlapped Schatten norm regularizes the rank of $X$ in terms of Tucker decomposition [16, 28]. Although the Tucker rank of $X^*$ is unknown in general, the convex optimization adjusts the rank depending on $\lambda_n$.

To solve the convex problem (3), the ADMM algorithm is often employed [1, 26, 28]. Since the overlapped Schatten norm is not differentiable, the ADMM algorithm avoids the differentiation of the regularization term by alternatively minimizing the augmented Lagrangian function iteratively.

## 3 Convex Formulation of TT Rank Minimization

To adopt TT decomposition to the convex optimization problem as (3), we need the convex surrogation of TT rank. For that purpose, we introduce the *Schatten TT norm* [22] as follows:

$$\|X\|_{s,T} := \frac{1}{K-1} \sum_{k=1}^{K-1} \|Q_k(X)\|_s := \frac{1}{K-1} \sum_{k=1}^{K-1} \sum_j \sigma_j(Q_k(X)), \tag{4}$$

where $Q_k : \mathcal{X} \to \mathbb{R}^{I_{\leq k} \times I_{k<}}$ is a reshaping operator that converts a tensor to a large matrix where the first $k$ modes are combined into the rows and the rest $K - k$ modes are combined into the columns. Oseledets *et al.*[21] shows that the matrix rank of $Q_k(X)$ can bound the $k$-th TT rank of $X$, implying that the Schatten TT norm surrogates the sum of the TT rank. Putting the Schatten TT norm into (3), we obtain the following optimization problem:

$$\min_{X \in \mathcal{X}} \left[ \frac{1}{2n} \|Y - \mathfrak{X}(X)\|^2 + \lambda_n \|X\|_{s,T} \right]. \tag{5}$$

### 3.1 ADMM Algorithm

To solve (5), we consider the augmented Lagrangian function $L(x, \{Z_k\}_{k=1}^{K-1}, \{\alpha_k\}_{k=1}^{K-1})$, where $x \in \mathbb{R}^{\prod_k I_k}$ is the vectorization of $X$, $Z_k$ is a reshaped matrices with size $I_{\leq k} \times I_{k<}$, and $\alpha_k \in \mathbb{R}^{\prod_k I_k}$ is a coefficient for constraints. Given initial points $(x^{(0)}, \{Z_k^{(0)}\}_k, \{\alpha_k^{(0)}\}_k)$, the $\ell$-th step of ADMM

is written as follows:

$$x^{(\ell+1)} = \left( \widetilde{\Omega}^T Y + n\eta \frac{1}{K-1} \sum_{k=1}^{K-1} (V_k(Z_k^{(\ell)}) - \alpha_k^{(\ell)}) \right) / (1 + n\eta K),$$

$$Z_k^{(\ell+1)} = \text{prox}_{\lambda_n/\eta}(V_k^{-1}(x^{(\ell+1)} + \alpha_k^{(\ell)})), \quad k = 1, \ldots, K,$$

$$\alpha_k^{(\ell+1)} = \alpha_k^{(\ell)} + (x^{(\ell+1)} - V_k(Z_k^{(\ell+1)})), \quad k = 1, \ldots, K.$$

Here, $\widetilde{\Omega}$ is an $n \times \prod_{k=1}^{I_k}$ matrix that works as the inversion mapping of $\mathfrak{X}$; $V_k$ is a vectorizing operator of an $I_{\leq k} \times I_{k<}$ matrix; $\text{prox}(\cdot)$ is the shrinkage operation of the singular values as $\text{prox}_b(W) = U \max\{S - bI, 0\}V^T$, where $USV^T$ is the singular value decomposition of $W$; $\eta > 0$ is a hyperparameter for a step size. We stop the iteration when the convergence criterion is satisfied (e.g. as suggested by Tomioka *et al.*[28]). Since the Schatten TT norm (4) is convex, the sequence of the variables of ADMM is guaranteed to converge to the optimal solution ([5, Theorem 5.1]). We refer to this algorithm as *TT-ADMM*.

TT-ADMM requires huge resources in terms of both time and space. For the time complexity, the proximal operation of the Schatten TT norm, namely the SVD thresholding of $V_k^{-1}$, yields the dominant complexity, which is $O(I^{3K/2})$ time. For the space complexity, we have $O(K)$ variables of size $O(I^K)$, which requires $O(KI^K)$ space.

## 4 Alternating Minimization with Randomization

In this section, we consider reducing the space complexity for handling higher order tensors. The idea is simple: we only maintain the TT format of the input tensor rather than the input tensor itself. This leads the following optimization problem:

$$\min_{\mathcal{G}} \left[ \frac{1}{2n} \|Y - \mathfrak{X}(X(\mathcal{G}))\|^2 + \lambda_n \|X(\mathcal{G})\|_{s,T} \right]. \tag{6}$$

Remember that $\mathcal{G} = \{G_k\}_k$ is the set of TT components and $X(\mathcal{G})$ is the tensor given by the TT format with $\mathcal{G}$. Now we only need to store the TT components $\mathcal{G}$, which drastically improves the space efficiency.

### 4.1 Randomized Schatten TT norm

We approximate the optimization of the Schatten TT norm. To avoid the computation of exponentially large-scale SVDs in the Schatten TT norm, we employ a technique called the "very sparse random projection" [12]. The main idea is that, if the size of a matrix is sufficiently larger than its rank, then its singular values (and vectors) are well preserved even after the projection by a sparse random matrix. This motivates us to use the Schatten TT norm over the random projection.

Preliminary, we introduce tensors for the random projection. Let $D_1, D_2 \in \mathbb{N}$ be the size of the matrix after projection. For each $k = 1, \ldots, K-1$ and parameters, let $\Pi_{k,1} \in \mathbb{R}^{D_1 \times I_1 \times \cdots \times I_k}$ be a tensor whose elements are independently and identically distributed as follows:

$$[\Pi_{k,1}]_{d_1,i_1,\ldots,i_k} = \begin{cases} +\sqrt{s/d_1} & \text{with probability } 1/2s, \\ 0 & \text{with probability } 1 - 1/s, \\ -\sqrt{s/d_1} & \text{with probability } 1/2s, \end{cases} \tag{7}$$

for $i_1, \ldots, i_k$ and $d_1 = 1, \ldots, D_1$. Here, $s > 0$ is a hyperparameter controlling sparsity. Similarly, we introduce a tensor $\Pi_{k,2} \in \mathbb{R}^{D_2 \times I_{k+1} \times \cdots \times I_{K-1}}$ that is defined in the same way as $\Pi_{k,1}$. With $\Pi_{k,1}$ and $\Pi_{k,2}$, let $\mathcal{P}_k : \mathcal{X} \to \mathbb{R}^{D_1 \times D_2}$ be a random projection operator whose element is defined as follows:

$$[\mathcal{P}_k(X)]_{d_1,d_2} = \sum_{j_1=1}^{I_1} \cdots \sum_{j_K=1}^{I_K} [\Pi_{k,1}]_{d_1,j_1,\ldots,j_k} [X]_{j_1,\ldots,j_K} [\Pi_{k,2}]_{d_2,j_{k+1},\ldots,j_K}. \tag{8}$$

Note that we can compute the above projection by using the facts that $X$ has the TT format and the projection matrices are sparse. Let $\pi_j^{(k)}$ be a set of indexes of non-zero elements of $\Pi_{k,j}$. Then, using the TT representation of $X$, (8) is rewritten as

$$[\mathcal{P}_k(X(\mathcal{G}))]_{d_1,d_2} = \sum_{(j_1,\ldots,j_k)\in\pi_1^{(k)}} [\Pi_{k,1}]_{d_1,j_1,\ldots,j_k}[G_1]_{j_1}\cdots[G_k]_{j_k}$$
$$\sum_{(j_{k+1},\ldots,j_K)\in\pi_2^{(k)}} [G_k]_{j_{k+1}}\cdots[G_K]_{j_K}[\Pi_{k,2}]_{d_2,j_{k+1},\ldots,j_K},$$

If the projection matrices have only $S$ nonzero elements (i.e., $S = |\pi_j^{(1)}| = |\pi_j^{(2)}|$), the computational cost of the above equation is $O(D_1 D_2 S K R^3)$.

The next theorem guarantees that the Schatten-1 norm of $\mathcal{P}_k(X)$ approximates the original one.

**Theorem 1.** *Suppose $X \in \mathcal{X}$ has TT rank $(R_1,\ldots,R_k)$. Consider the reshaping operator $Q_k$ in (4), and the random operator $\mathcal{P}_k$ as (8) with tensors $\Pi_{k,1}$ and $\Pi_{k,2}$ defined as (7). If $D_1, D_2 \geq \max\{R_k, 4(\log(6R_k) + \log(1/\epsilon))/\epsilon^2\}$, and all the singular vectors $u$ of $Q(X)_k$ are well-spread as $\sum_j |u_j|^3 \leq \epsilon/(1.6k\sqrt{s})$, we have*

$$\frac{1-\epsilon}{R_k}\|Q_k(X)\|_s \leq \|\mathcal{P}_k(X)\|_s \leq (1+\epsilon)\|Q_k(X)\|_s,$$

*with probability at least $1 - \epsilon$.*

Note that the well-spread condition can be seen as a stronger version of the incoherence assumption which will be discussed later.

## 4.2   Alternating Minimization

Note that the new problem (6) is non-convex because $X(\mathcal{G})$ does not form a convex set on $\mathcal{X}$. However, if we fix $\mathcal{G}$ except for $G_k$, it becomes convex with respect to $G_k$. Combining with the random projection, we obtain the following minimization problem:

$$\min_{G_k} \left[ \frac{1}{2n}\|Y - \mathfrak{X}(X(\mathcal{G}))\|^2 + \frac{\lambda_n}{K-1}\sum_{k'=1}^{K-1}\|\mathcal{P}_{k'}(X(\mathcal{G}))\|_s \right]. \tag{9}$$

We solve this by the ADMM method for each $k = 1,\ldots,K$. Let $g_k \in \mathbb{R}^{I_k R_{k-1} R_k}$ be the vectorization of $G_k$, and $W_{k'} \in \mathbb{R}^{D_1 \times D_2}$ be a matrix for the randomly projected matrix. The augmented Lagrangian function is then given by $L_k(g_k, \{W_{k'}\}_{k'=1}^{K-1}, \{\beta_{k'}\}_{k'=1}^{K-1})$, where $\{\beta_{k'} \in \mathbb{R}^{D_1 D_2}\}_{k'=1}^{K-1}$ are the Lagrange multipliers. Starting from initial points $(g_k^{(0)}, \{W_{k'}^{(0)}\}_{k'=1}^{K-1}, \{\beta_{k'}^{(0)}\}_{k'=1}^{K-1})$, the $\ell$-th ADMM step is written as follows:

$$g_k^{(\ell+1)} = \left( \Omega^T\Omega/n + \eta\sum_{k'=1}^{K-1}\Gamma_{k'}^T\Gamma_{k'} \right)^{-1} \left( \Omega^TY/n + \frac{1}{K-1}\sum_{k'=1}^{K-1}\Gamma_{k'}^T(\eta\widetilde{V}_k(W_{k'}^{(\ell)}) - \beta_{k'}^{(\ell)}) \right),$$

$$W_{k'}^{(\ell+1)} = \text{prox}_{\lambda_n/\eta}\left( \widetilde{V}_k^{-1}(\Gamma_{k'}g_k^{(\ell+1)} + \beta_{k'}^{(\ell)}) \right), \quad k' = 1,\ldots,K-1,$$

$$\beta_{k'}^{(\ell+1)} = \beta_{k'}^{(\ell)} + (\Gamma_{k'}g_k^{(\ell+1)} - \widetilde{V}_k(W_{k'}^{(\ell+1)})), \quad k' = 1,\ldots,K-1.$$

Here, $\Gamma^{(k)} \in \mathbb{R}^{D_1 D_2 \times I_k R_{k-1} R_k}$ is the matrix imitating the mapping $G_k \mapsto \mathcal{P}_k(X(G_k; \mathcal{G}\backslash\{G_k\}))$, $\widetilde{V}_k$ is a vectorizing operator of $D_1 \times D_2$ matrix, and $\Omega$ is an $n \times I_k R_{k-1} R_k$ matrix of the operator $\mathfrak{X} \circ X(\cdot; \mathcal{G}\backslash\{G_k\})$ with respect to $g_k$. Similarly to the convex approach, we iterate the ADMM steps until convergence. We refer to this algorithm as *TT-RAM*, where RAM stands for randomized least square.

The time complexity of TT-RAM at the $\ell$-th iteration is $O((n + KD^2)KI^2R^4)$; the details are deferred to Supplementary material. The space complexity is $O(n + KI^2R^4)$, where $O(n)$ is for $Y$ and $O(KI^2R^4)$ is for the parameters.

# 5 Theoretical Analysis

In this section, we investigate how the TT rank and the number of observations affect to the estimation error. Note that all the proofs of this section are deferred to Supplementary material.

## 5.1 Convex Solution

To analyze the statistical error of the convex problem (5), we assume the *incoherence* of the reshaped version of $X^*$.

**Assumption 2.** *(Incoherence Assumption) There exists $k \in \{1, \ldots, K\}$ such that a matrix $Q_k(X^*)$ has orthogonal singular vectors $\{u_r \in \mathbb{R}^{I_{\leq k}}, v_r \in \mathbb{R}^{I_{k<}}\}_{r=1}^{R_k}$ satisfying*

$$\max_{1 \leq i \leq I_{<k}} \|P_U e_i\| \leq (\mu R_k / I_{\leq k})^{\frac{1}{2}} \quad and \quad \max_{1 \leq i \leq I_{<k}} \|P_V e_i\| \leq (\mu R_k / I_{k<})^{\frac{1}{2}}$$

*with some $0 \leq \mu < 1$. Here, $P_U$ and $P_V$ are linear projections onto spaces spanned by $\{u_r\}_r$ and $\{v_r\}_r$; $\{e_i\}_i$ is the natural basis.*

Intuitively, the incoherence assumption requires that the singular vectors for the matrix $Q_k(X^*)$ are well separated. This type of assumption is commonly used in the matrix and tensor completion studies [2, 3, 31]. Under the incoherence assumption, the error rate of the solution of (5) is derived.

**Theorem 3.** *Let $X^* \in \mathcal{X}$ be a true tensor with TT rank $(R_1, \ldots, R_{K-1})$, and let $\widehat{X} \in \mathcal{X}$ be the minimizer of (3). Suppose that $\lambda_n \geq \|\mathfrak{X}^*(\mathcal{E})\|_\infty / n$ and that Assumption 2 for some $k' \in \{1, 2, \ldots, K\}$ is satisfied. If*

$$n \geq C_{m'} \mu_{k'}^2 \max\{I_{\leq k'}, I_{k'<}\} R_{k'} \log^3 \max\{I_{\leq k'}, I_{k'<}\}$$

*with a constant $C_{m'}$, then with probability at least $1 - (\max\{I_{\leq k'}, I_{k'<}\})^{-3}$ and with a constant $C_X$,*

$$\|\widehat{X} - X^*\|_F \leq C_X \frac{\lambda_n}{K} \sum_{k=1}^{K-1} \sqrt{R_k}.$$

Theorem 3 states that the bound for the statistical error gets larger as the TT rank increases. In other words, completing a tensor is relatively easy as long as the tensor has small TT rank. Also, when we set $\lambda_n \to 0$ as $n$ increases, we can state the consistency of the minimizer.

The result of Theorem 3 is similar to that obtained from the studies on matrix completion [3, 16] and tensor completion with the Tucker decomposition or SVD [28, 31]. Note that, although Theorem 3 is for tensor completion, the result can easily be generalized to other settings such as the tensor recovery or the compressed sensing problems.

## 5.2 TT-RAM Solution

Prior to the analysis, let $\mathcal{G}^*$ be the true TT components such that $X^* = X(\mathcal{G}^*)$. For simplification, we assume that the elements of $\mathcal{G}^*$ are normalized, i.e., $\|G_k\| = 1, \forall k$, and an $R_k \times I_{k-1} I_k$ matrix reshaped from $G_k^*$ has a $R_k$ row rank.

In addition to the incoherence property (Assumption 2), we introduce an additional assumption on the initial point of the ALS iteration.

**Assumption 4.** *(Initial Point Assumption) Let $\mathcal{G}^{init} := \{G_k^{init}\}_{k=1}^K$ be the initial point of the ALS iteration procedure. Then, there exists a finite constant $C_\gamma$ that satisfies*

$$\max_{k \in \{1, \ldots, K\}} \|G_k^{init} - G_k^*\|_F \leq C_\gamma.$$

Assumption 4 requires that the initial point is sufficiently close to the true solutions $\mathcal{G}^*$. Although the ALS method is not guaranteed to converge to the global optimum in general, Assumption 4 guarantees the convergence to the true solutions [25]. Now we can evaluate the error rate of the solution obtained by TT-RAM.

**Theorem 5.** *Let $X(\mathcal{G}^*)$ be the true tensor generated by $\mathcal{G}^*$ with TT rank $(R_1, \ldots, R_{K-1})$, and $\widehat{\mathcal{G}} = \mathcal{G}^t$ be the solution of TT-RAM at the $t$-th iteration. Further, suppose that Assumption 2 for some $k' \in \{1, 2, \ldots, K\}$ and Assumption 4 are satisfied, and suppose that Theorem (1) holds with $\epsilon > 0$ for $k = 1, \ldots, K$. Let $C_m, C_A, C_B > 0$ be $0 < \chi < 1$ be some constants. If*

$$n \geq C_m \mu_{k'}^2 R_{k'} \max\{I_{\leq k'}, I_{k'<}\} \log^3 \max\{I_{\leq k'}, I_{k'<}\},$$

*and the number of iterations $t$ satisfies $t \geq (\log \chi)^{-1}\{\log(C_B \lambda_n K^{-1}(1+\epsilon) \sum_k \sqrt{R_k}) - \log C_\gamma\}$, then with probability at least $1 - \epsilon(\max\{I_{\leq k'}, I_{k'<}\})^{-3}$ and for $\lambda_n \geq \|\mathfrak{X}^*(\mathcal{E})\|_\infty/n$,*

$$\|X(\widehat{\mathcal{G}}) - X(\mathcal{G}^*)\|_F \leq C_A(1+\epsilon)\lambda_n \sum_{k=1}^{K-1} \sqrt{R_k}. \tag{10}$$

Again, we can obtain the consistency of TT-RAM by setting $\lambda_n \to 0$ as $n$ increases. Since the setting of $\lambda_n$ corresponds to that of Theorem 3, the speed of convergence of TT-RAM in terms of $n$ is equivalent to the speed of TT-ADMM.

By comparing with the convex approach (Theorem 3), the error rate becomes slightly worse. Here, the term $\lambda_n \sum_{k=1}^{K-1} \sqrt{R_k}$ in (10) comes from the estimation by the alternating minimization, which linearly increases by $K$. This is because there are $K$ optimization problems and their errors are accumulated to the final solution. The term $(1 + \epsilon)$ in (10) comes from the random projection. The size of the error $\epsilon$ can be arbitrary small by controlling the parameters of the random projection $D_1, D_2$ and $s$.

## 6  Related Work

To solve the tensor completion problem with TT decomposition, Wang *et al.*[30] and Grasedyck *et al.*[6] developed algorithms that iteratively solve minimization problems with respect to $G_k$ for each $k = 1, \ldots, K$. Unfortunately, the adaptivity of the TT rank is not well discussed. [30] assumed that the TT rank is given. Grasedyck *et al.*[6] proposed a grid search method. However, the TT rank is determined by a single parameter (i.e., $R_1 = \cdots = R_{K-1}$) and the search method lacks its generality. Furthermore, the scalability problem remains in both methods—they require more than $O(I^K)$ space.

Phien et al. [22] proposed a convex optimization method using the Schatten TT norm. However, because they employed an alternating-type optimization method, the global convergence of their method is not guaranteed. Moreover, since they maintain $X$ directly and perform the reshape of $X$ several times, their method requires $O(I^K)$ time.

Table 1 highlights the difference between the existing and our methods. We emphasize that our study is the first attempt to analyze the statistical performance of TT decomposition. In addition, TT-RAM is only the method that both time and space complexities do not grow exponentially in $K$.

| Method | Global Convergence | Rank Adaptivity | Time Complexity | Space Complexity | Statistical Bounds |
|---|---|---|---|---|---|
| TCAM-TT[30] | | | $O(nIKR^4)$ | $O(I^K)$ | |
| ADF for TT[6] | | (search) | $O(KIR^3 + nKR^2)$ | $O(I^K)$ | |
| SiLRTC-TT[22] | | ✓ | $O(I^{3K/2})$ | $O(KI^K)$ | |
| **TT-ADMM** | ✓ | ✓ | $O(KI^{3K/2})$ | $O(I^K)$ | ✓ |
| **TT-RAM** | | ✓ | $O((n + KD^2)KI^2R^4)$ | $O(n + KI^2R^4)$ | ✓ |

Table 1: Comparison of TT completion algorithms, with $R$ is a parameter for the TT rank such that $R = R_1 = \cdots = R_{K-1}$, $I = I_1 = \cdots = I_K$ is dimension, $K$ is the number of modes, $n$ is the number of observed elements, and $D$ is the dimension of random projection.

## 7  Experiments

### 7.1  Validation of Statistical Efficiency

Using synthetic data, we verify the theoretical bounds derived in Theorems 3 and 5. We first generate TT components $\mathcal{G}^*$; each component $G_k^*$ is generated as $G_k^* = G_k^\dagger/\|G_k^\dagger\|_F$ where each

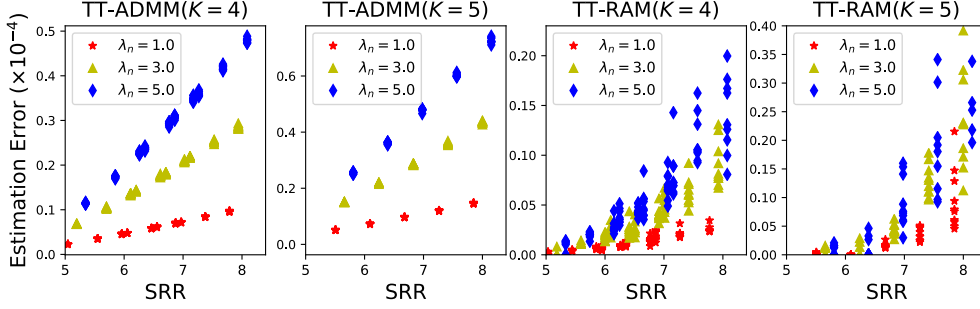

Figure 1: Synthetic data: the estimation error $\|\widehat{X} - X^*\|_F$ against SRR $\sum_k \sqrt{R_k}$ with the order-4 tensor $(K = 4)$ and the order-5 tensor $(K = 5)$. For each rank and $\lambda_n$, we measure the error by 10 trials with different random seeds, which affect both the missing pattern and the initial points.

Table 2: Electricity data: the prediction error and the runtime (in seconds).

| Method | $K = 5$ | | $K = 7$ | | $K = 8$ | | $K = 10$ | |
| --- | --- | --- | --- | --- | --- | --- | --- | --- |
| | Error | Time | Error | Time | Error | Time | Error | Time |
| Tucker | 0.219 | 7.125 | 0.371 | 610.61 | N/A | N/A | N/A | N/A |
| TCAM-TT | 0.219 | 2.174 | 0.928 | 27.497 | 0.928 | 146.651 | N/A | N/A |
| ADF for TT | 0.998 | 1.221 | 1.160 | 23.211 | 1.180 | 278.712 | N/A | N/A |
| SiLRTC-TT | 0.339 | 1.478 | 0.928 | 206.708 | N/A | N/A | N/A | N/A |
| TT-ADMM | 0.221 | 0.289 | 1.019 | 154.991 | 1.061 | 2418.00 | N/A | N/A |
| TT-RAM | 0.219 | 4.644 | 0.928 | 4.726 | 0.928 | 7.654 | 1.173 | 7.968 |

element of $G_k^\dagger$ is sampled from the i.i.d. standard normal distribution. Then we generate $Y$ by following the generative model (2) with the observation ratio $n / \prod_k I_k = 0.5$ and the noise variance 0.01. We prepare two tensors of different size: an order-4 tensor of size $8 \times 8 \times 10 \times 10$ and an order-5 tensor of size $5 \times 5 \times 7 \times 7 \times 7$. At the order-4 tensor, the TT rank is set as $(R_1, R_2, R_3)$ where $R_1, R_2, R_3 \in \{3, 5, 7\}$. At the order-5 tensor, the TT rank is set as $(R_1, R_2, R_3, R_4)$ where $R_1, R_2, R_3, R_4 \in \{2, 4\}$. For estimation, we set the size of $G_k$ and $\Pi_k$ as 10, which is larger than the true TT rank. The regularization coefficient $\lambda_n$ is selected from $\{1, 3, 5\}$. The parameters for random projection are set as $s = 20$ and $D_1 = D_2 = 10$.

Figure 1 shows the relation between the estimation error and the sum of root rank (SRR) $\sum_k \sqrt{R_k}$. The result of TT-ADMM shows that the empirical errors are linearly related to SSR which is shown by the theoretical result. The result of TT-RAM roughly replicates the theoretical relationship.

## 7.2 Higher-Order Markov Chain for Electricity Data

We apply the proposed tensor completion methods for analyzing the electricity consumption data [13]. The dataset contains time series measurements of household electric power consumption for every minutes from December 2006 to November 2010 and it contains over $200,000$ observations.

The higher-order Markov chain is a suitable method to represent long-term dependency, and it is a common tool of time-series analysis [7] and natural language processing [9]. Let $\{W_t\}_t$ be discrete-time random variables take values in a finite set $B$, and the order-$K$ Markov chain describes the conditional distribution of $W_t$ with given $\{W_\tau\}_{\tau < t}$ as $P(W_t | \{W_\tau\}_{\tau < t}) = P(W_t | W_{t-1}, \ldots, W_{t-K})$. As $K$ increases, the conditional distribution of $W_t$ can include more information from the past observations.

We complete the missing values of $K$-th Markov transition of the electricity dataset. We discretize the value of the dataset into 10 values and set $K \in \{5, 7, 8, 10\}$. Next, we empirically estimate the conditional distribution of size $10^K$ using $200,000$ observations. Then, we create $X$ by randomly selecting $n = 10,000$ elements from the the conditional distribution and regarding the other elements as missing. After completion, the prediction error is measured. We select hyperparameters using a grid search with cross-validation.

Figure 2 compares the prediction error and the runtime by the related studies with TT decomposition. For reference, we also report those values by Tucker decomposition without TT. When $K = 5$, the rank adaptive methods achieve low estimation errors. As $K$ increases, however, all the methods except for TT-RAM suffers from the scalability issue. Indeed, at $K = 10$, only TT-RAM works and the others does not due to exhausting memory.

## 8    Conclusion

In this paper, we investigated TT decomposition from the statistical and computational viewpoints. To analyze its statistical performance, we formulated the convex tensor completion problem via the low-rank TT decomposition using the TT Schatten norm. In addition, because the optimization of the convex problem is infeasible, we developed an alternative algorithm called TT-RAM by combining with the ideas of random projection and alternating minimization. Based on this, we derived the error bounds of estimation for both the convex minimizer and the solution obtained by TT-RAM. The experiments supported our theoretical results and demonstrated the scalability of TT-RAM.

## Acknowledgement

We thank Prof. Taiji Suzuki for comments that greatly improved the manuscript. M. Imaizumi is supported by Grant-in-Aid for JSPS Research Fellow (15J10206) from the JSPS. K. Hayashi is supported by ONR N62909-17-1-2138.

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
