[Supplementary Material]

# Supplementary Material for
## "On Tensor Train Rank Minimization:
## Statistical Efficiency and Scalable Algorithm"

## A   Proof of Theorem 1

The theorem is obtained immediately by combining Li et al. [12] and Mu et al. [15].

Let $\Pi : \mathbb{R}^n \to \mathbb{R}^k$ be a sparse random projection defined by

$$\Pi_{ij} = \begin{cases} +\sqrt{s/k} & \text{probability } 1/2s, \\ 0 & \text{probability } 1 - 1/s, \\ -\sqrt{s/k} & \text{probability } 1/2s. \end{cases} \tag{11}$$

Then the following theorem holds.

**Lemma 6** (Lemma 4 in [12]). *Let $u$ be a unit vector. Then $\sqrt{k}(\Pi u)_i \to N(0,1)$ and $k\|\Pi u\|_2^2 \to \chi_k^2$ in law with the convergence rate*

$$|P(\sqrt{k}(\Pi u)_i < t) - P(N(0,1) < t)| \leq 0.8\sqrt{s}\sum_j |u_j|^3. \tag{12}$$

$\square$

Suppose that $\sum_j |u_j|^3 \leq e^{-k\epsilon^2/4}/(1.6k\sqrt{s})$. Then

$$|P(\|\Pi u\|_2^2 \in [1-\epsilon, 1+\epsilon]) - P(\chi_k^2 \in [k(1-\epsilon), k(1+\epsilon)])| \leq e^{-k\epsilon^2/4}. \tag{13}$$

By using

$$P(\chi_k^2 \in [k(1-\epsilon), k(1+\epsilon)]) \leq 2e^{-k(\epsilon^2-\epsilon^3)/4}, \tag{14}$$

we have

$$P(\|\Pi u\|_2^2 \in [1-\epsilon, 1+\epsilon]) \leq 3e^{-k(\epsilon^2-\epsilon^3)/4}. \tag{15}$$

The preservation of $L_2$ norm implies the preservation of the Schatten-1 norm as follows.

**Lemma 7** (Restatement of Theorem 1 in [15]). *Let $Z$ be an $m \times n$ matrix with rank $r$. If $k \geq r$ and $\|\Pi u\|_2^2 \in [1-\epsilon, 1+\epsilon]$ for all singular vectors $u$ of $Z$, we have*

$$\sqrt{(1-\epsilon)/r}\|Z\|_s \leq \|\Pi Z\|_s \leq \sqrt{1+\epsilon}\|Z\|_s. \tag{16}$$

$\square$

Now we prove Theorem 1. Let $Z = Q_k(X)$ be a $\prod_{k'=1}^{k} I_{k'} \times \prod_{k'=k+1}^{K} I_{k'}$ matrix obtained by reshaping tensor $X$. Since $X$ is a TT of rank $(R_1, \ldots, R_K)$, the rank of matrix $Z$ is at most $R_k$. By applying Lemma 7 twice, we obtain

$$\frac{1-\epsilon}{R_k}\|Z\|_s \leq \|\Pi Z\|_s \leq (1+\epsilon)\|Z\|_s \tag{17}$$

with some probability. If $\sum_j |u_j|^3 \leq e^{-k\epsilon^2/4}/(1.6k\sqrt{s})$ for all singular vectors of $Z$, the probability is at least $1 - 6R_k e^{-k\epsilon^2/4}$ since there are $2R_k$ singular vectors.

## B   Proof of Theorem 3

*Proof.* Since $\widehat{X}$ is the minimizer of the optimization problem, we have the following basic inequality

$$\frac{1}{2n}\|Y - \mathfrak{X}(\widehat{X})\|^2 + \lambda_n \sum_{k=1}^{K-1} \|Q_k(\widehat{X})\|_s \leq \frac{1}{2n}\|Y - \mathfrak{X}(\widehat{X}^*)\|^2 + \lambda_n \sum_{k=1}^{K-1} \|Q_k(X^*)\|_s.$$

Using the relation that

$$\|Y - \mathfrak{X}(\widehat{X})\|^2 = \|(Y - \mathfrak{X}(X^*)) - (\mathfrak{X}(\widehat{X}) - \mathfrak{X}(X^*))\|^2$$
$$= \|Y - \mathfrak{X}(X^*)\|^2 + \|\mathfrak{X}(\widehat{X}) - \mathfrak{X}(X^*)\|^2 - 2\langle Y - \mathfrak{X}(X^*), \mathfrak{X}(\widehat{X}) - \mathfrak{X}(X^*)\rangle,$$

we rewrite the basic inequality as

$$\frac{1}{2n}\|\mathfrak{X}(\widehat{X}) - \mathfrak{X}(X^*)\|^2 \le \frac{1}{n}\langle\mathfrak{X}(\widehat{X}) - \mathfrak{X}(X^*), \mathcal{E}\rangle + \lambda_n \sum_{k=1}^{K-1}\left(\|Q_k(X^*)\|_s - \|Q_k(\widehat{X})\|_s\right).$$

Define the error $\Delta := \widehat{X} - X^*$. Applying the Hölder's inequality, we have

$$\frac{1}{n}\langle\mathfrak{X}(\Delta), \mathcal{E}\rangle = \frac{1}{n}\langle\Delta, \mathfrak{X}^*(\mathcal{E})\rangle = \frac{1}{n}\frac{1}{K-1}\sum_{k=1}^{K-1}\langle Q_k(\Delta), \mathfrak{X}^*(\mathcal{E})\rangle$$

$$\le \frac{1}{n}\frac{1}{K-1}\sum_{k=1}^{K-1}\|\mathcal{E}\|_\infty\|Q_k(\Delta)\|_s \le \frac{\lambda_n}{K-1}\sum_{k=1}^{K-1}\|Q_k(\Delta)\|_s,$$

where $\mathfrak{X}^*$ is an adjoint operator of $\mathfrak{X}$ and the last inequality holds by the setting of $\lambda_n$. Also, the triangle inequality and the linearity of $Q_k(\cdot)$ yield

$$\|Q_k(X^*)\|_s - \|Q_k(\widehat{X})\|_s \le \|Q_k(\Delta)\|_s.$$

Then, we bound the inequality as

$$\frac{1}{2n}\|\mathfrak{X}(\Delta)\|^2 \le \frac{\lambda_n}{K-1}\sum_{k=1}^{K-1}\|Q_k(\Delta)\|_s + \frac{\lambda_n}{K-1}\sum_{k=1}^{K-1}\|Q_k(\Delta)\|_s = \frac{2\lambda_n}{K-1}\sum_{k=1}^{K-1}\|Q_k(\Delta)\|_s.$$

To bound $\|Q_k(\Delta)\|_s$, we apply the result of Lemma 1 in [16] and Lemma 2 in [28]. Along with proof of the lemmas, we obtain the property that a rank of $Q_k(\Delta)$ is bounded by $2R_k$, thus the Cauchy-Schwartz inequality implies

$$\|Q_k(\Delta)\|_s \le \sqrt{2R_k}\|Q_k(\Delta)\|_F.$$

Then we obtain

$$\|\mathfrak{X}(\Delta)\|_F^2 \le \frac{2\lambda_n}{K-1}\sum_{k=1}^{K-1}\sqrt{2R_k}\|Q_k(\Delta)\|_F.$$

We apply the completion theory by [3] and [2] to bound $\|\mathfrak{X}(\Delta)\|_F^2$ below. Let $k' \in \{1, \ldots, K\}$ be the index which satisfies Assumption 2, and we have $\|\mathfrak{X}(\Delta)\|^2 = \|\widetilde{\mathfrak{X}}(Q_{k'}(\Delta))\|^2$ where $\widetilde{\mathfrak{X}}$ is a rearranging operator for the reshaped tensor. Then, Theorem 7 in [3] yields that

$$\|Q_{k'}(\Delta)\|_F \le \left(\sqrt{\frac{48\min\{I_{\le k'}, I_{k'<}\}}{n}} + 1\right)\|\widetilde{\mathfrak{X}}(Q_{k'}(\Delta))\|,$$

with probability at least $1 - (\max\{I_{\le k'}, I_{k'<}\})^{-3}$ and

$$n \ge C_{m'}\mu_{k'}^2 \max\{I_{\le k'}, I_{k'<}\}R_{k'}\log^3\max\{I_{\le k'}, I_{k'<}\},$$

with a constant $C_{m'} > 0$. Then we obtain that

$$\frac{1}{n}\|\widetilde{\mathfrak{X}}(Q_{k'}(\Delta))\|^2 \ge C_{k'}\|Q_{k'}(\Delta)\|_F^2 = C_{k'}\|\Delta\|_F^2,$$

where $C_\kappa = (144\min\{I_{\le k'}, I_{k'<}\} + 3n)^{-1} > 0$.

Finally, we have

$$\|\Delta\|_F^2 \le C_{\kappa'}^{-1}\frac{2\lambda_n}{K-1}\sum_{k=1}^{K-1}\sqrt{2R_k}\|Q_k(\Delta)\|_F$$

$$= C_{\kappa'}^{-1}\|\Delta\|_F\frac{2\lambda_n}{K-1}\sum_{k=1}^{K-1}\sqrt{2R_k}$$

$$= 3C_{\kappa'}^{-1}\|\Delta\|_F\frac{\lambda_n}{K}\sum_{k=1}^{K-1}\sqrt{2R_k}.$$

Dividing both hands side by $\|\Delta\|_F$ provides the result.

$\square$

## C    Proof of Theorem 5

Preliminarily, we introduce an alternative formation of the optimization problem. For each $k \in \{1, 2, \ldots, K\}$, we rewrite the term $X_k(\mathcal{G})$ as

$$G_k \times_2 G_{<k} \times_3 G_{k<},$$

where $\times_j$ denotes the $j$-mode product (for detail, see [10]). Here, $G_{<k}$ is a tensor with size $R_k \times I_1 \times \cdots \times I_{K-1}$ and its element is given as

$$[G_{<k}]_{r,j_1,\ldots,j_{k-1}} = [G_1]_{j_1,:,:}[G_2]_{j_2,:,:} \cdots [G_{k-1}]_{j_{k-1},:,r},$$

for $j_{k'} = 1, \ldots, I_{k'}$ and $r = 1, \ldots, R_k$. Namely, $G_{<k}$ is the left side of tensor train decomposition of $X$ than $G_k$. Similarly, $G_{k<}$ is a tensor with size $R_{k+1} \times I_{k+1} \times \cdots \times I_K$ and its element is given as

$$[G_{k<}]_{r,j_{k+1},\ldots,j_K} = [G_{k+1}]_{j_{k+1},r,:} \cdots [G_K]_{j_K,:,:},$$

for $j_{k'} = 1, \ldots, I_{k'}$ and $r = 1, \ldots, R_{k+1}$. Using the result, the ALS optimization problem (9) is rewritten as

$$\min_{G_k} \left[ \frac{1}{2n} \|Y - \mathfrak{X}(G_k \times_2 G_{<k} \times_3 G_{k<})\|^2 + \frac{\lambda_n}{K-1} \sum_{k'=1}^{K-1} \|\mathcal{P}_{k'}(G_k \times_2 G_{<k} \times_3 G_{k<})\|_s \right]. \quad (18)$$

When $k = 1$, we set $G_{<k} = 1$. Similarly, when $k = K$, $G_{k<} = 1$ holds.

Using the formula, we investigate the convergence of $G_k$ by fixing other elements as $G_{<k} = \widetilde{G}_{<k}$ and $G_{k<} = \widetilde{G}_{k<}$. Let $\{G_k^*\}_{k^*=1}^K$ be a set of tensor which formulates the true tensor $X^*$. Also, $G_{<k}^*$ and $G_{k<}^*$ are defined similarly. To evaluate the convergence, we introduce that

$$\Xi(\widetilde{\mathcal{G}}) := \max_{k \in \{1,\ldots,K\}} \left[ \|\widetilde{G}_k - G_k^*\|_F \right].$$

We obtain the following lemma which evaluates the optimization of (18) with given $\widetilde{G}_{<k}$ and $\widetilde{G}_{k<}$.

**Lemma 8.** *For each $k \in \{1, \ldots, K\}$, consider the optimization* (18) *with respect to $G_k$ with given $\widetilde{\mathcal{G}}$. Then, with probability at least $1 - (\max\{I_{\leq k'}, I_{k'<}\})^{-3}$ and*

$$n \geq C_m \mu_{k'}^2 \max\{I_{\leq k'}, I_{k'<}\} R_{k'} \log^n \max\{I_{\leq k'}, I_{k'<}\},$$

*we obtain*

$$\|\widehat{G}_k - G_k^*\| \leq 6(C_\kappa C_K)^{-1} \left\{ 2(K-1)C_K(1 + n^{-1})\Xi(\widetilde{\mathcal{G}}) + \frac{2\lambda_n(2+\epsilon)}{K-1} \sum_{k'=1}^{K-1} \sqrt{2R_{k'}} \right\}.$$

*Proof.* Our proof takes following four steps: 1) derive a basic inequality from the optimality condition, 2) bound terms of the RHS of the basis inequality, 3) bound below the LHS of the basis inequality, and 4) combine the result.

**Step 1.**    Derive a basic inequality.

By the optimality condition of (18) with given $\widetilde{G}_{<k}$ and $\widetilde{G}_{k<}$, we have

$$\frac{1}{2n} \|Y - \mathfrak{X}(\widehat{G}_k \times_2 \widetilde{G}_{<k} \times_3 \widetilde{G}_{k<})\|^2 + \frac{\lambda_n}{K-1} \sum_{k'=1}^{K-1} \|\mathcal{P}_{k'}(\widehat{G}_k \times_2 \widetilde{G}_{<k} \times_3 \widetilde{G}_{k<})\|_s$$

$$\leq \frac{1}{2n} \|Y - \mathfrak{X}(G_k^* \times_2 \widetilde{G}_{<k} \times_3 \widetilde{G}_{k<})\|^2 + \frac{\lambda_n}{K-1} \sum_{k'=1}^{K-1} \|\mathcal{P}_{k'}(G_k^* \times_2 \widetilde{G}_{<k} \times_3 \widetilde{G}_{k<})\|_s. \quad (19)$$

Using the triangle inequality and the linearity of $\mathfrak{X}$ and the mode product $\times_j$, we obtain

$$\|Y - \mathfrak{X}(\widehat{G}_k \times_2 \widetilde{G}_{<k} \times_3 \widetilde{G}_{k<})\|^2$$
$$= \|\{Y - \mathfrak{X}(G_k^* \times_2 \widetilde{G}_{<k} \times_3 \widetilde{G}_{k<})\} - \{\mathfrak{X}(\widehat{G}_k \times_2 \widetilde{G}_{<k} \times_3 \widetilde{G}_{k<}) - \mathfrak{X}(G_k^* \times_2 \widetilde{G}_{<k} \times_3 \widetilde{G}_{k<})\}\|^2$$
$$= \|\{Y - \mathfrak{X}(G_k^* \times_2 \widetilde{G}_{<k} \times_3 \widetilde{G}_{k<})\} - \mathfrak{X}((\widehat{G}_k - G_k^*) \times_2 \widetilde{G}_{<k} \times_3 \widetilde{G}_{k<})\|^2$$
$$= \|Y - \mathfrak{X}(G_k^* \times_2 \widetilde{G}_{<k} \times_3 \widetilde{G}_{k<})\|^2 + \|\mathfrak{X}((\widehat{G}_k - G_k^*) \times_2 \widetilde{G}_{<k} \times_3 \widetilde{G}_{k<})\|^2$$
$$- 2\langle Y - \mathfrak{X}(G_k^* \times_2 \widetilde{G}_{<k} \times_3 \widetilde{G}_{k<}), \mathfrak{X}((\widehat{G}_k - G_k^*) \times_2 \widetilde{G}_{<k} \times_3 \widetilde{G}_{k<})\rangle.$$

Substituting the result into (19), we obtain that

$$\frac{1}{2n}\|\mathfrak{X}((\widehat{G}_k - G_k^*) \times_2 \widetilde{G}_{<k} \times_3 \widetilde{G}_{k<})\|^2 + \frac{\lambda_n}{K-1}\sum_{k'=1}^{K-1}\|\mathcal{P}_{k'}(\widehat{G}_k \times_2 \widetilde{G}_{<k} \times_3 \widetilde{G}_{k<})\|_s$$
$$\leq \frac{1}{n}\langle Y - \mathfrak{X}(G_k^* \times_2 \widetilde{G}_{<k} \times_3 \widetilde{G}_{k<}), \mathfrak{X}((\widehat{G}_k - G_k^*) \times_2 \widetilde{G}_{<k} \times_3 \widetilde{G}_{k<})\rangle$$
$$+ \frac{\lambda_n}{K-1}\sum_{k'=1}^{K-1}\|\mathcal{P}_{k'}(G_k^* \times_2 \widetilde{G}_{<k} \times_3 \widetilde{G}_{k<})\|_s. \tag{20}$$

About the regularization term, we apply the following inequality

$$\frac{\lambda_n}{K-1}\sum_{k'=1}^{K-1}\|\mathcal{P}_{k'}(G_k^* \times_2 \widetilde{G}_{<k} \times_3 \widetilde{G}_{k<})\|_s - \frac{\lambda_n}{K-1}\sum_{k'=1}^{K-1}\|\mathcal{P}_{k'}(\widehat{G}_k \times_2 \widetilde{G}_{<k} \times_3 \widetilde{G}_{k<})\|_s$$
$$\leq \frac{\lambda_n}{K-1}\sum_{k'=1}^{K-1}\|\mathcal{P}_{k'}((G_k^* - \widehat{G}_k) \times_2 \widetilde{G}_{<k} \times_3 \widetilde{G}_{k<})\|_s$$
$$= \frac{\lambda_n}{K-1}\sum_{k'=1}^{K-1}\Big(\|Q_{k'}((G_k^* - \widehat{G}_k) \times_2 \widetilde{G}_{<k} \times_3 \widetilde{G}_{k<})\|_s$$
$$- \|\mathcal{P}_{k'}((G_k^* - \widehat{G}_k) \times_2 \widetilde{G}_{<k} \times_3 \widetilde{G}_{k<})\|_s - \|Q_{k'}((G_k^* - \widehat{G}_k) \times_2 \widetilde{G}_{<k} \times_3 \widetilde{G}_{k<})\|_s\Big), \tag{21}$$

by using the triangle inequality and the linearity of the random projection operator $\mathcal{P}_{k'}$. Here, we apply Theorem 1 with $\epsilon$ and obtain

$$\left(\frac{1-\epsilon}{2R_{k'}} - 1\right)\|Q_{k'}((G_k^* - \widehat{G}_k) \times_2 \widetilde{G}_{<k} \times_3 \widetilde{G}_{k<})\|_s$$
$$\leq \|\mathcal{P}_{k'}((G_k^* - \widehat{G}_k) \times_2 \widetilde{G}_{<k} \times_3 \widetilde{G}_{k<})\|_s - \|Q_{k'}((G_k^* - \widehat{G}_k) \times_2 \widetilde{G}_{<k} \times_3 \widetilde{G}_{k<})\|_s$$
$$\leq \epsilon\|Q_{k'}((G_k^* - \widehat{G}_k) \times_2 \widetilde{G}_{<k} \times_3 \widetilde{G}_{k<})\|_s.$$

Here, the denominator in the left hand side follows Lemma 1 in [16]. Then we have

$$\left|\|\mathcal{P}_{k'}((G_k^* - \widehat{G}_k) \times_2 \widetilde{G}_{<k} \times_3 \widetilde{G}_{k<})\|_s - \|Q_{k'}((G_k^* - \widehat{G}_k) \times_2 \widetilde{G}_{<k} \times_3 \widetilde{G}_{k<})\|_s\right|$$
$$\leq \max\{\epsilon, |(1-\epsilon)/(2R_{k'}) - 1|\}\|Q_{k'}((G_k^* - \widehat{G}_k) \times_2 \widetilde{G}_{<k} \times_3 \widetilde{G}_{k<})\|_s$$
$$\leq (1+\epsilon)\|Q_{k'}((G_k^* - \widehat{G}_k) \times_2 \widetilde{G}_{<k} \times_3 \widetilde{G}_{k<})\|_s.$$

Using this result and continue (21) then we have

$$(21) \leq \frac{\lambda_n}{K-1}\sum_{k'=1}^{K-1}(2+\epsilon)\|Q_{k'}((G_k^* - \widehat{G}_k) \times_2 \widetilde{G}_{<k} \times_3 \widetilde{G}_{k<})\|_s. \tag{22}$$

About the first term of the RHS of (20), we decompose it as

$$\frac{1}{n}\langle Y - \mathfrak{X}(G_k^* \times_2 \widetilde{G}_{<k} \times_3 \widetilde{G}_{k<}), \mathfrak{X}((\widehat{G}_k - G_k^*) \times_2 \widetilde{G}_{<k} \times_3 \widetilde{G}_{k<})\rangle$$

$$= \frac{1}{n}\langle Y - \mathfrak{X}(G_k^* \times_2 G_{<k}^* \times_3 G_{k<}^*), \mathfrak{X}((\widehat{G}_k - G_k^*) \times_2 \widetilde{G}_{<k} \times_3 \widetilde{G}_{k<})\rangle$$

$$+ \frac{1}{n}\langle \mathfrak{X}(G_k^* \times_2 (\widetilde{G}_{<k} - G_{<k}^*) \times_3 G_{k<}^*), \mathfrak{X}((\widehat{G}_k - G_k^*) \times_2 \widetilde{G}_{<k} \times_3 \widetilde{G}_{k<})\rangle$$

$$+ \frac{1}{n}\langle \mathfrak{X}(G_k^* \times_2 G_{<k}^* \times_3 (\widetilde{G}_{k<} - G_{k<}^*)), \mathfrak{X}((\widehat{G}_k - G_k^*) \times_2 \widetilde{G}_{<k} \times_3 \widetilde{G}_{k<})\rangle$$

$$+ \frac{1}{n}\langle \mathfrak{X}(G_k^* \times_2 (\widetilde{G}_{<k} - G_{<k}^*) \times_3 (\widetilde{G}_{k<} - G_{k<}^*)), \mathfrak{X}((\widehat{G}_k - G_k^*) \times_2 \widetilde{G}_{<k} \times_3 \widetilde{G}_{k<})\rangle$$

$$=: T_0 + T_1 + T_2 + T_3.$$

About the term $T_0$, we use the observation model (2) and the adjoint operator $\mathfrak{X}^*$ then obtain

$$T_0 = \frac{1}{n}\langle \mathcal{E}, \mathfrak{X}((\widehat{G}_k - G_k^*) \times_2 \widetilde{G}_{<k} \times_3 \widetilde{G}_{k<})\rangle$$

$$= \frac{1}{n}\langle \mathfrak{X}^*(\mathcal{E}), (\widehat{G}_k - G_k^*) \times_2 \widetilde{G}_{<k} \times_3 \widetilde{G}_{k<}\rangle.$$

Since the reshaping does not affect the value of the inner product, we continue to evaluate $T_0$ as

$$T_0 = \frac{1}{n}\langle \mathfrak{X}^*(\mathcal{E}), (\widehat{G}_k - G_k^*) \times_2 \widetilde{G}_{<k} \times_3 \widetilde{G}_{k<}\rangle$$

$$= \frac{1}{n(K-1)}\sum_{k'=1}^{K-1}\langle Q_{k'}(\mathfrak{X}^*(\mathcal{E})), Q_{k'}((\widehat{G}_k - G_k^*) \times_2 \widetilde{G}_{<k} \times_3 \widetilde{G}_{k<})\rangle$$

$$\leq \frac{1}{n(K-1)}\sum_{k'=1}^{K-1}\|Q_{k'}(\mathfrak{X}^*(\mathcal{E}))\|_\infty \|Q_{k'}((\widehat{G}_k - G_k^*) \times_2 \widetilde{G}_{<k} \times_3 \widetilde{G}_{k<})\|_s$$

$$= \frac{1}{n(K-1)}\|\mathfrak{X}^*(\mathcal{E})\|_\infty \sum_{k'=1}^{K-1}\|Q_{k'}((\widehat{G}_k - G_k^*) \times_2 \widetilde{G}_{<k} \times_3 \widetilde{G}_{k<})\|_s$$

$$\leq \frac{\lambda_n}{(K-1)}\sum_{k'=1}^{K-1}\|Q_{k'}((\widehat{G}_k - G_k^*) \times_2 \widetilde{G}_{<k} \times_3 \widetilde{G}_{k<})\|_s.$$

The first inequality follows the Hölder's inequality, and the second inequality is derived by the setting of $\lambda_n$.

Substituting (22) and the bounds with $T_1, T_2, T_3$ and $T_0$ into (20), finally we obtain

$$\frac{1}{2n}\|\mathfrak{X}((\widehat{G}_k - G_k^*) \times_2 \widetilde{G}_{<k} \times_3 \widetilde{G}_{k<})\|^2$$

$$\leq T_1 + T_2 + T_3 + \underbrace{\frac{\lambda_n}{K-1}\sum_{k'=1}^{K-1}(2+\epsilon)\|Q_{k'}((G_k^* - \widehat{G}_k) \times_2 \widetilde{G}_{<k} \times_3 \widetilde{G}_{k<})\|_s}_{=:T_4}. \qquad (23)$$

Here, we obtain the basic inequality.

**Step 2.**  Bound the RHS of the basic inequality.

For brevity, we introduce notation

$$\widetilde{\Delta}_k := (\widehat{G}_k - G_k^*) \times_2 \widetilde{G}_{<k} \times_3 \widetilde{G}_{k<}.$$

We bound $T_1$ by using the Cauchy-Schwartz inequality as

$$T_1 = \frac{1}{n}\langle \mathfrak{X}(G_k^* \times_2 (\widetilde{G}_{<k} - G_{<k}^*) \times_3 G_{k<}^*), \mathfrak{X}((\widehat{G}_k - G_k^*) \times_2 \widetilde{G}_{<k} \times_3 \widetilde{G}_{k<})\rangle$$

$$\leq \frac{1}{n}\|\mathfrak{X}(G_k^* \times_2 (\widetilde{G}_{<k} - G_{<k}^*) \times_3 G_{k<}^*)\|\|\mathfrak{X}((\widehat{G}_k - G_k^*) \times_2 \widetilde{G}_{<k} \times_3 \widetilde{G}_{k<})\|$$

$$\leq \frac{1}{n}\|G_k^* \times_2 (\widetilde{G}_{<k} - G_{<k}^*) \times_3 G_{k<}^*\|_F\|(\widehat{G}_k - G_k^*) \times_2 \widetilde{G}_{<k} \times_3 \widetilde{G}_{k<}\|_F$$

$$\leq \frac{1}{n}\|G_k^* \times_2 (\widetilde{G}_{<k} - G_{<k}^*) \times_3 G_{k<}^*\|_F\|\widetilde{\Delta}_k\|_F,$$

here we use the relation $\|\mathfrak{X}(X)\|^2 \leq \|X\|_F^2$ for all $X \in \Theta$. We introduce a constant $c_k$ for $k = 1, \ldots, K$ which satisfying $c_k \geq \|A \times_k G_k^*\|_F/\|A\|_F$ where $A$ is a tensor with proper size. Since we suppose that the reshaped matrix from $G_k^*$ has $R_k$ row rank, we can guarantee that $c_k$ is positive and finite. Using $c_k$, we have

$$\|G_k^* \times_2 (\widetilde{G}_{<k} - G_{<k}^*) \times_3 G_{k<}^*\|$$

$$\leq c_k \prod_{k'>k} c_{k'}\|\widetilde{G}_{<k} - G_{<k}^*\|_F$$

$$\leq c_k \prod_{k'>k} c_{k'} \sum_{k'<k} \|\widetilde{G}_{k'} - G_{k'}^*\|_F \prod_{\ell<k,\ell\neq k'} c_\ell$$

$$\leq \prod_{k'\geq k} c_{k'}(k-1) \prod_{\ell<k} c_\ell \Xi(\widetilde{\mathcal{G}})$$

$$= (k-1)\prod_{k'=1}^{K} c_{k'}\Xi(\widetilde{\mathcal{G}}).$$

Here, we define $C_K := \prod_{k'=1}^{K} c_{k'}$, we obtain

$$T_1 \leq \frac{1}{n}(k-1)C_K\Xi(\widetilde{\mathcal{G}}). \tag{24}$$

Similarly, we obtain

$$T_2 \leq \frac{1}{n}(K-k)C_K\Xi(\widetilde{\mathcal{G}}). \tag{25}$$

About $T_3$, we have

$$T_3 = \frac{1}{n}\langle \mathfrak{X}(G_k^* \times_2 (\widetilde{G}_{<k} - G_{<k}^*) \times_3 (\widetilde{G}_{k<} - G_{k<}^*)), \mathfrak{X}((\widehat{G}_k - G_k^*) \times_2 \widetilde{G}_{<k} \times_3 \widetilde{G}_{k<})\rangle$$

$$\leq \frac{1}{n}\|G_k^* \times_2 (\widetilde{G}_{<k} - G_{<k}^*) \times_3 (\widetilde{G}_{k<} - G_{k<}^*)\|_F\|\Delta_k\|_F.$$

We evaluate the first norm as

$$\|G_k^* \times_2 (\widetilde{G}_{<k} - G_{<k}^*) \times_3 (\widetilde{G}_{k<} - G_{k<}^*)\|_F$$

$$\leq c_k \left(\|\widetilde{G}_{<k} \times_3 (\widetilde{G}_{k<} - G_{k<}^*)\|_F + \|G_{<k}^* \times_3 (\widetilde{G}_{k<} - G_{k<}^*)\|_F\right)$$

$$\leq \frac{2}{n}(K-1)C_K\Xi(\widetilde{\mathcal{G}}).$$

Then, we have

$$T_3 \leq \frac{2}{n}(K-1)C_K\Xi(\widetilde{\mathcal{G}}). \tag{26}$$

To bound $T_4$, we apply the same line of the proof of Theorem 3. Along with Lemma 1 in [16], we bound the Schatten norm of $Q_k(\widetilde{\Delta}_k)$ and apply the Cauchy-Schwartz inequality, then obtain

$$T_4 \leq \frac{2\lambda_n(2+\epsilon)}{K-1}\sum_{k'=1}^{K-1}\sqrt{2R_{k'}}\|Q_{k'}(\widetilde{\Delta}_k)\| = \frac{2\lambda_n(2+\epsilon)}{K-1}\sum_{k'=1}^{K-1}\sqrt{2R_{k'}}\|\widetilde{\Delta}_k\|.$$

Combining the bound with (24), (25) and (26), we update the bound (23) as

$$\frac{1}{2n}\|\mathfrak{X}(\widetilde{\Delta}_k)\|^2 \leq \frac{3(K-1)C_K}{n}\Xi(\widetilde{\mathcal{G}})\|\mathfrak{X}(\widetilde{\Delta}_k)\| + \frac{2\lambda_n(2+\epsilon)}{K-1}\sum_{k'=1}^{K-1}\sqrt{2R_{k'}}\|\widetilde{\Delta}_k\|$$

$$\leq \frac{3(K-1)C_K}{n}\Xi(\widetilde{\mathcal{G}})\|\widetilde{\Delta}_k\|_F + \frac{2\lambda_n(2+\epsilon)}{K-1}\sum_{k'=1}^{K-1}\sqrt{2R_{k'}}\|\widetilde{\Delta}_k\|. \qquad (27)$$

**Step 3.**   Bound below the LHS of the basic inequality.

We apply the matrix completion theory developed by [3] and [2]. Let $k' \in \{1, \ldots, K\}$ be the index satisfying Assumption 2. Since the value of the $L^2$-norm and the Frobenius norm is invariant to the shape of tensors, we compare the value of $\|Q_{k'}(\Delta_k)\|_F$ and $\|\widetilde{\mathfrak{X}}(Q_{k'}(\Delta_k))\|_F$ with $k'$ instead of $\|\Delta_k\|_F$ and $\|\mathfrak{X}(\Delta_k)\|_F$.

For the matrix $Q_{k'}(X^*)$, we apply Assumption 2 and obtain that $Q_{k'}(X^*)$ has the $\mu_{k'}$-incoherence property. Then, we apply Theorem 2 and Theorem 7 in [3], we obtain the following inequality as

$$\|Q_{k'}(\widetilde{\Delta}_k)\|_F \leq \left(\sqrt{\frac{48\min\{I_{\leq k'}, I_{k'<}\}}{n}} + 1\right)\|\mathfrak{X}(Q_{k'}(\widetilde{\Delta}_k))\|,$$

with probability at least $1 - (\max\{I_{\leq k'}, I_{k'<}\})^{-3}$ and

$$n \geq C_m\mu_{k'}^2\max\{I_{\leq k'}, I_{k'<}\}R_{k'}\log^3\max\{I_{\leq k'}, I_{k'<}\},$$

with a constant $C_m > 0$. Then we obtain that

$$\frac{1}{n}\|\mathfrak{X}(\widetilde{\Delta}_k)\|^2 = \frac{1}{n}\|\widetilde{\mathfrak{X}}(Q_{k'}(\widetilde{\Delta}_k))\|^2$$

$$\geq (144\min\{I_{\leq k'}, I_{k'<}\} + 3n)^{-1}\|Q_{k'}(\widetilde{\Delta}_k)\|_F^2 =: C_\kappa\|Q_{k'}(\widetilde{\Delta}_k)\|_F^2 = C_\kappa\|\widetilde{\Delta}_k\|_F^2,$$

where $C_\kappa > 0$ since $n \leq \prod_k I_k$. Using this result into (27), we have

$$\frac{C_\kappa}{6}\|\widetilde{\Delta}_k\|_F^2 \leq \frac{3(K-1)C_K}{n}\Xi(\widetilde{\mathcal{G}})\|\widetilde{\Delta}_k\|_F + \frac{2\lambda_n(2+\epsilon)}{K-1}\sum_{k'=1}^{K-1}\sqrt{2R_{k'}}\|\widetilde{\Delta}_k\|_F.$$

Then we obtain the inequality

$$\frac{C_\kappa}{6}\|\widetilde{\Delta}_k\|_F^2 \leq \frac{3(K-1)C_K}{n}\Xi(\widetilde{\mathcal{G}})\|\widetilde{\Delta}_k\|_F + \frac{2\lambda_n(2+\epsilon)}{K-1}\sum_{k'=1}^{K-1}\sqrt{2R_{k'}}\|\widetilde{\Delta}_k\|_F.$$

We divide the both hands side by $\|\widetilde{\Delta}_k\|_F$ about the first term, and consider the root about the second term, then we have

$$\frac{C_\kappa}{6}\|\widetilde{\Delta}_k\|_F \leq \frac{3(K-1)C_K}{n}\Xi(\widetilde{\mathcal{G}}) + \frac{2\lambda_n(2+\epsilon)}{K-1}\sum_{k'=1}^{K-1}\sqrt{2R_{k'}}, \qquad (28)$$

by using the property $\|\mathfrak{X}(X)\| \leq \|X\|$ for all $X \in \mathcal{X}$.

Finally, we define

$$\Delta_k := (\widehat{G}_k - G_k^*) \times_2 G_{<k}^* \times_3 G_{k<}^*,$$

and compare $\Delta_k$ and $\widetilde{\Delta}_k$ as

$$\|\Delta_k\| \leq \|\widetilde{\Delta}_k\| + \|\widetilde{\Delta}_k - \Delta_k\|.$$

We evaluate the last term by the same way of the step 2 as

$$\|\widetilde{\Delta}_k - \Delta_k\|_F$$

$$\leq \left\|(\widehat{G}_k - G_k^*) \times_2 (G_{<k}^* \times_3 G_{k<}^* - \widetilde{G}_{<k} \times_3 \widetilde{G}_{k<})\right\|_F$$

$$\leq 2c_k\left\{\left\|(G_{<k}^* - \widetilde{G}_{<k}) \times_3 G_{k<}^*\right\|_F + \left\|\widetilde{G}_{<k} \times_3 (G_{k<}^* - \widetilde{G}_{k<})\right\|_F\right\}$$

$$\leq 2(K-1)C_K\Xi(\widetilde{\mathcal{G}}).$$

Then, we have

$$\|\Delta_k\|_F - 2(K-1)C_k\Xi(\widetilde{\mathcal{G}}) \leq \|\widetilde{\Delta}_k\|_F.$$

Substituting the result into (28), we obtain

$$\frac{C_\kappa}{6}\|\Delta_k\|_F \leq 2(K-1)C_K(1+n^{-1})\Xi(\widetilde{\mathcal{G}}) + \frac{2\lambda_n(2+\epsilon)}{K-1}\sum_{k'=1}^{K-1}\sqrt{2R_{k'}}. \qquad (29)$$

**Step 4.** Combining the results.

Substituting the result of the step 3 into (29), we finally obtain

$$\|\widehat{G}_k - G_k^*\| \leq 6(C_\kappa C_K)^{-1}2(K-1)C_K(1+n^{-1})\Xi(\widetilde{\mathcal{G}}) + \frac{2\lambda_n(2+\epsilon)}{K-1}\sum_{k'=1}^{K-1}\sqrt{2R_{k'}}.$$

$$\square$$

We back to the proof of Theorem 5. Based on the result of Lemma 8, we will take two steps: (a) evaluate the distance between $X(\widetilde{\mathcal{G}})$ and $X(\mathcal{G}^*)$, and (b) show the convergence as the ALS iteration proceeds.

**Step (a).** Evaluate the distance between $X(\widetilde{\mathcal{G}})$ and $X(\mathcal{G}^*)$.

For brevity, we introduce new notation for $X(\mathcal{G})$. Using the tensor product, we denote

$$X(\mathcal{G}) = G_1 \times_2 G_2 \times_3 \cdots \times_{K-1} G_{K-1} \times_K G_K.$$

Then, we evaluate the distance between $X(\mathcal{G})$ and $X(\mathcal{G}^*)$ as

$$\begin{aligned}
&X(\widetilde{\mathcal{G}}) - X(\mathcal{G}^*) \\
&= \widetilde{G}_1 \times_2 \cdots \times_K \widetilde{G}_K - G_1^* \times_2 \cdots \times_K G_K^* \\
&= (\widetilde{G}_1 \times_2 \cdots \times_{K-1} \widetilde{G}_{K-1} \times_K \widetilde{G}_K - \widetilde{G}_1 \times_2 \cdots \times_{K-1} \widetilde{G}_{K-1} \times_K G_K^*) \\
&\quad + (\widetilde{G}_1 \times_2 \cdots \times_{K-1} \widetilde{G}_{K-1} \times_K G_K^* - \widetilde{G}_1 \times_2 \cdots \times_{K-1} G_{K-1}^* \times_K G_K^*) \\
&\cdots \\
&\quad + (\widetilde{G}_1 \times_2 G_2^* \times_3 \cdots \times_{K-1} G_{K-1}^* \times_K G_K^* - G_1^* \times_2 \cdots \times_K G_K^*) \\
&= \sum_{k=1}^{K} \widetilde{G}_{<k} \times_k (\widetilde{G}_k - G_k^*) \times_{k+1} G_{k<}^*.
\end{aligned}$$

Then, we consider the Frobenius norm as

$$\begin{aligned}
&\|X(\widetilde{\mathcal{G}}) - X(\mathcal{G}^*)\|_F^2 \\
&= \left\|\sum_{k=1}^{K} \widetilde{G}_{<k} \times_k (\widetilde{G}_k - G_k^*) \times_{k+1} G_{k<}^*\right\|_F^2 \\
&= \sum_{k=1}^{K}\sum_{k'=1}^{K} \left\langle \widetilde{G}_{<k} \times_k (\widetilde{G}_k - G_k^*) \times_{k+1} G_{k<}^*, \widetilde{G}_{<k'} \times_{k'} (\widetilde{G}_{k'} - G_{k'}^*) \times_{k'+1} G_{k'<}^* \right\rangle \\
&= \sum_{k=1}^{K} \left\|\widetilde{G}_{<k} \times_k (\widetilde{G}_k - G_k^*) \times_{k+1} G_{k<}^*\right\|_F^2 \\
&\quad + \sum_{k=1}^{K}\sum_{k'\neq k} \left\langle \widetilde{G}_{<k} \times_k (\widetilde{G}_k - G_k^*) \times_{k+1} G_{k<}^*, \widetilde{G}_{<k'} \times_{k'} (\widetilde{G}_{k'} - G_{k'}^*) \times_{k'+1} G_{k'<}^* \right\rangle.
\end{aligned}$$

As same as the proof of Lemma 8, we bound the first term as

$$\left\| \widetilde{G}_{<k} \times_k (\widetilde{G}_k - G_k^*) \times_{k+1} \widetilde{G}_{k<} \right\|_F^2 \le C_K^2 \Xi^2(\widetilde{\mathcal{G}}).$$

For the second term, we obtain

$$\left\langle \widetilde{G}_{<k} \times_k (\widetilde{G}_k - G_k^*) \times_{k+1} \widetilde{G}_{k<}, \widetilde{G}_{<k'} \times_{k'} (\widetilde{G}_{k'} - G_{k'}^*) \times_{k'+1} \widetilde{G}_{k'<} \right\rangle$$

$$\le \left\| \widetilde{G}_{<k} \times_k (\widetilde{G}_k - G_k^*) \times_{k+1} \widetilde{G}_{k<} \right\|_F \left\| \widetilde{G}_{<k'} \times_{k'} (\widetilde{G}_{k'} - G_{k'}^*) \times_{k'+1} \widetilde{G}_{k'<} \right\|_F$$

$$\le C_K^2 \Xi^2(\widetilde{\mathcal{G}}).$$

Combining the results, we obtain

$$\|X(\widetilde{\mathcal{G}}) - X(\mathcal{G}^*)\|_F^2 \le (K + K^2) C_k^2 \Xi^2(\widetilde{\mathcal{G}}).$$

**Step (b).**   Show convergence with the ALS iteration.

Let $\mathcal{G}^t$ be a set $\mathcal{G}$ obtained by $t$-th ALS iteration. By the result of the step (a), we have

$$\|X(\mathcal{G}^t) - X(\mathcal{G}^*)\|_F^2 \le (K + K^2) C_K^2 \Xi(\mathcal{G}^t).$$

Applying the result of Lemma 8, let $\widehat{G}_k^t$ be the minimizer of optimization of (18) with the $t$-th ALS iteration, we obtain for each $t = 1, 2, \ldots$,

$$\Xi(\mathcal{G}^t) = \max_k \|\widehat{G}_k^t - G_k^*\|_F \le 6(C_\kappa C_K)^{-1} 2(K-1) C_K (1 + n^{-1}) \Xi(\widetilde{\mathcal{G}}) + \frac{2\lambda_n(2+\epsilon)}{K-1} \sum_{k'=1}^{K-1} \sqrt{2R_{k'}}.$$

The inequality holds since $\mathcal{G}^{t-1}$ is the fixed $\widetilde{\mathcal{G}}$ for the $t$-th ALS iteration. We define the contraction coefficient

$$\chi := 12 C_\kappa^{-1} (K-1) C_K (1.5 + n^{-1}),$$

and using the assumption that $\chi < 1$, we have

$$\Xi(\mathcal{G}^t) \le \max \left\{ \chi^t \Xi(\mathcal{G}^0), 6(C_\kappa C_K)^{-1} \frac{2\lambda_n(2+\epsilon)}{K} \sum_{k'=1}^{K-1} \sqrt{2R_{k'}} \right\}, \tag{30}$$

where $\mathcal{G}^0$ is an initial value of $\mathcal{G}$. With Assumption 4, we set $t$ sufficiently large as

$$t \ge (\log \chi)^{-1} \left\{ \log \left( 6(C_\kappa C_K)^{-1} \left( \frac{2\lambda_n(2+\epsilon)}{K} \sum_{k'=1}^{K-1} \sqrt{2R_{k'}} \right) \right) - \log \Xi(\mathcal{G}^0) \right\},$$

we obtain

$$\|X(\mathcal{G}^t) - X(\mathcal{G}^*)\|_F^2 \le 12 C_\kappa^{-1} K^2 C_K \left( \frac{2\lambda_n(2+\epsilon)}{K} \sum_{k'=1}^{K-1} \sqrt{2R_{k'}} \right)^2.$$

As we set $\widehat{X} := X(\mathcal{G}^t)$, we obtain the result.

$\square$

# D   Time Complexity of TT-RAM

To update $g_k^{(\ell+1)}$, we need to compute

- $A = \Omega^T \Omega$, which requires $O(nI^2 R^4)$,
- $B = \sum_{k'=1}^{K-1} \Gamma_{k'}^T \Gamma_{k'}$, which requires $O(KD^2 I^2 R^4)$,
- the inversion of an $IR^2 \times IR^2$ matrix $(A + B)$, which requires $O(I^3 R^6)$,

- $c = \Omega^T Y$, which requires $O(nIR^2)$,
- $d = \widetilde{V}_k(W_{k'}^{(\ell)})$, which requires $O(D^2)$,
- $e = \frac{1}{K-1}\sum_{k'=1}^{K-1}\Gamma_{k'}^T(\eta d - \beta_{k'}^{(\ell)}$, which requires $O(KD^2IR^2)$,

To update $W_{k'}^{(\ell+1)}$, we need to compute

- $a = \Gamma_{k'}g_k^{(\ell+1)}$, which requires $O(D^2IR^2)$,
- $\widetilde{V}_k^{-1}(a + \beta_{k'}^{(\ell)})$, which requires $O(D^2)$,
- the proximal operation, which requires $O(D^3)$.

To update $\beta_{k'}^{(\ell+1)}$, we need to compute

- $a = \Gamma_{k'}g_k^{(\ell+1)}$, which requires $O(D^2IR^2)$,
- $b = \widetilde{V}_k(W_{k'}^{(\ell+1)})$, which requires $O(D^2)$,

Because there are $g_k^{(\ell+1)}$ for $k = 1,\ldots,K$, $W_{k'}^{(\ell+1)}$ for $k,k' = 1,\ldots,K$, and $\beta_{k'}^{(\ell+1)}$ for $k,k' = 1,\ldots,K$, the total time complexity is

$$
\begin{aligned}
&O(K(nI^2R^4 + KD^2I^2R^4 + I^3R^6 + nIR^2 + D^2 + KD^2IR^2)) \\
&\quad + O(K^2(D^2IR^2 + D^2 + D^3)) + O(K^2(D^2IR^2 + D^2)) \\
&= O(K(nI^2R^4 + KD^2I^2R^4 + I^3R^6)) + O(K^2(D^2IR^2 + D^3)) \\
&= O(K(nI^2R^4 + KD^2I^2R^4 + I^3R^6)) + O(K^2D^2IR^2) \\
&= O(nKI^2R^4 + K^2D^2I^2R^4 + KI^3R^6) \\
&= O(nKI^2R^4 + K^2D^2I^2R^4)
\end{aligned}
$$

In the third line and the last line, we assumed $D = O(IR^2)$ and $IR^2 = O(n)$, respectively.