[Reviews · NeurIPS 2017]

Reviewer 1



The authors propose two algorithms for fitting low-rank tensor-train (TT) decompositions using the Schatten TT norm as the low-rank inducing regularizer. The first (TT-ADMM) uses ADMM with the optimization variable being the tensor itself--- this has exponential space and time complexities. The second (TT-RALS) reduces these complexities to polynomial by optimizing in an alternating fashion (using ADMM) over the factors in a TT factorization of the tensor, and using randomized dimensionality reduction to estimate the Schatten TT norm. Theoretical analysis is provided to argue that: under some regularity conditions (incoherence) the matrix estimate obtained by TT-ADMM is consistent, and if the initial guess is close enough to the optimal point, TT-RALS is also consistent. Quantitative rates of convergence are provided. I verified the correctness of the first claim. Experimental results are given comparing the performance of TT-ADMM and TT-RALS to other TT low-rank factorization methods for matrix recovery. These show that TT-RALS is much more scalable than the other algorithms. TT factorizations are useful and preferable over other forms of tensor factorizations precisely because they allow you to express high-order tensors with a tractable number of parameters, but the challenge is in learning these factorizations efficiently. TT-RALS is a very useful algorithm to that end, and the theoretical guarantees given in this paper are worth dissemination. -I suggest the authors change the name from TT-RALS since that suggest alternating least squares, which is not what the algorithm does. - TT-ADMM is orders of magnitude more accurate on the synthetic data than TT-RALS, which is not surprising ... but on the example with real data, TT-RALS is the most accurate method, consistently beating TT-ADMM. Please comment on this counterintuitive result (is this due, e.g., to termination conditions?) - Please do another round of careful proof reading: there are several grammatically incorrect phrases like "the all" instead of "all the".

Reviewer 2



This paper looks at the tensor train (TT) decomposition as an efficient storage method for tensors with statistical guarantees. Traditional methods for tensor decomposition include Tucker decomposition [29] and CP decomposition. The advantage of TT decomposition is that it can avoid the exponential space complexity of Tucker decomposition. However, its statistical performance is less understood, which the authors attempt to address in this paper. In addition, the authors provide an efficient alternative to existing TT algorithms. It’s interesting to note that Tucker and TT decompositions have complementary bottleneck - One has space complexity issues and the second has computational complexity issues. The authors look at the problem of tensor completion which is an extension of the matrix completion problem that has been extensively researched in the literature. In matrix completion problem, the standard assumption made is that the matrix is low-rank so that the problem becomes well-posed. This assumption is also observed to be the case in practical applications such as recommender system. The authors make a similar observation for the tensor to be estimated (i.e. a low-rank tensor). Just as nuclear norm/schatten-p norm is used a relaxation to the rank matrix, the authors propose the Schatten TT norm as a convex relaxation of the TT rank of a tensor. Comments: - The authors mention that the TT rank is a tuple in section 2, while mentioning later in section 6 that it can be described by a single parameter, making the definition confusing. - The problem formulation (4), the algorithmic approach (section 3.1) all mimick the ideas in matrix completion literature. The innovation is more to do with the appropriate representation of tensors while extending these ideas from the matrix completion literature. - Even in the matrix factorization literature, the scalable methods are the ones that already assume a low-rank factor structure for the matrix to be completed to save space and computation. For tensors, it is all the more imperative to assume a low-rank factor structure. The authors do this in (6) to save space and time. - Theorem 1 is quite interesting and provides a more tractable surrogate for the TT schatten norm. - The statistical results are a new contribution for tensor completion, although the assumption of incoherence (Assumption 2) and subsequent proofs are by now very standard in the M-estimation literature. - The authors use an ADMM algorithm again for solving (6) in section 4.2. It is not clear this would have a better advantage as compared to just the simpler alternating minimization over the TT factors of the tensor. It is not clear if the convergence of ADMM to a local minima for non-convex problems with multiple factors is proven in the literature. The authors don’t cite a paper on this either. On the other hand, it is easy to see that the simpler alternating minimization (keeping other factors fixed) converges to a local optimum or a critical point. The authors don’t discuss any of these details in 4.2. - The number of samples needed for statistical convergence is in the worst-case the the product of the dimensions of the tensor. Perhaps a better result would be to show that the sample complexity is proportional to the true TT rank of the tensor? The authors don’t mention any discussion on the lower-bound on the sample complexity in the paper, making the goodness of the sample complexity bound in Theorem 5 questionable. Overall, the paper presents an interesting investigation of the TT decomposition for tensors and uses it in the context of tensor completion with statistical guarantees. It is not clear if the algorithm used for alternating optimization (ADMM) is actually guaranteed to converge though it might work in practice. Also, the authors don’t discuss the goodness (what’s a good lower bound?) of the sample complexity bounds derived for their algorithm for tensor completion. The approach to modeling, algorithm development, scalability and statistical convergence mimicks that taken for matrix completion - However, the authors do a good job of combining it all together and putting their approach in perspective with respective to existing algorithms in the literature.

Reviewer 3



This paper studies tensor completion problem under tensor-train(TT) format. It offers scalable algorithms as well as theoretical guarantees on recovery performance. However, I wish to hear some justifications from the authors on below problems. Currently, its quality is not good enough for acceptance to NIPS. 1). How about the comparison on recovery guarantee with other tensor formats? Such as Turker/CP decomposition, overlapping/latent tensor nuclear norm and t-SVD. From my point, with so many tensor formats at hand, it is not good to limit the scope of the submission only to TT format. Since the authors claim they are the first to give the statistical guarantee, thus I wish to see comparisons on statistical guarantees offered by different tensor formats. 2). While the paper targets at large scale problems, it can not really handle large tensor with lots of missing values. This is summarised in Table 1, where both time and space depend on the size of the tensor rather than the number of observations. It is not a problem for matrix completion problems [a], not a problem for tensor completion using Turker decomposition [d] or latent nuclear norm [e]. 3). A better method than random projection provided in Section 4.1 is power method. It has been successfully applied to matrix completion problem with nuclear norm regularisation [b]. So, Theorem 1 is not useful here. 4). There is no convergence guarantee for TT-RALS from optimisation's perspective. It is hard to check Assumption 2 and 4 in practice, this may suggestion that TT-RALS algorithm is not useful. 5). Besides, we can drop the TT-norm and directly optimise each factor in TT-format, this leads to a coordinate descent/alternative minimization algorithm (see [c,6,30]). For such algorithms, we always have convergence guarantee. The authors argue that for those methods, determination of TT-rank is a problem. However, the proposed method suffers similar problems, as they need to determine D and lambda_i. The author should include such algorithms in experiments for a comparison. 6). I would like to see a comparison of proposed method and other state-of-the-art methods. e.g. [d,e], on 3-order tensor. Then, it is more clear for readers to understand the difference on recovery performance induced by different tensor format. It is more interesting to see TT-format with proposed method beat those state-of-the-arts. 7). Could the authors perform other experiments in Section 7.2? The results in Table 2 suggests higher order tensor is not useful, the performance significantly deteriorates as the tensor order gets higher. 8). The authors may want to explain more on the novelty inside the proof. Reference [a]. Spectral Regularization Algorithms for Learning Large Incomplete Matrices. JMLR 2010 [b]. Nuclear Norm Minimization via Active Subspace Selection. ICML 2014 [c]. Tensorizing Neural Networks. NIPS 2015 [d]. Low-rank tensor completion: A Riemannian manifold preconditioning approach. ICML 2016 [e]. Efficient Sparse Low-Rank Tensor Completion using the Frank-Wolfe Algorithm. AAAI 2017